# The cost of noise: Stochastic punishment falls short of sustaining cooperation in social dilemma experiments

**Mohammad Salahshour, Vincent Oberhauser, Matteo Smerlak** *

Max Planck Institute for Mathematics in the Sciences, Leipzig, Germany

* smerlak@mis.mpg.de

## Abstract

Identifying mechanisms able to sustain costly cooperation among self-interested agents is a central problem across social and biological sciences. One possible solution is peer punishment: when agents have an opportunity to sanction defectors, classical behavioral experiments suggest that cooperation can take root. Overlooked from standard experimental designs, however, is the fact that real-world human punishment—the administration of justice—is intrinsically noisy. Here we show that stochastic punishment falls short of sustaining cooperation in the repeated public good game. As punishment noise increases, we find that contributions decrease and punishment efforts intensify, resulting in a 45% drop in gains compared to a noiseless control. Moreover, we observe that uncertainty causes a rise in antisocial punishment, a mutually harmful behavior previously associated with societies with a weak rule of law. Our approach brings to light challenges to cooperation that cannot be explained by economic rationality and strengthens the case for further investigations of the effect of noise—and not just bias—on human behavior.

## Introduction

The success of the group often requires individuals to cooperate, but cooperation is vulnerable to selfish incentives to defect [1, 2]. This tension—the defining feature of social dilemma—arises wherever public goods are involved [3]: if others bear the costs of maintaining a resource that anyone can access freely, then each agent is better off reaping its returns without also contributing. Given this selective pressure for selfishness, how can the tragedy of the commons [4], where all agents defect and everyone loses, be averted? Answering this question would be a key step towards understanding how human civilization came about—and how it might persist in the future.

In the last decades, prosocial (or altruistic) punishment [5, 6] has emerged as a possible solution to social dilemma. Prosocial punishment is a primitive form of self-governance [7] in which cooperators punish defectors to maximize group welfare, without the need for external enforcement mechanisms. Whether this strategy can evolve spontaneously is not obvious: since it is costly, punishment itself subject to free-riding. Nevertheless, punishment is

**Data Availability Statement:** All data will be made available after acceptance.

**Funding:** Funding for this work was provided by the Alexander von Humboldt Foundation in the framework of the Sofja Kovalevskaja Award endowed by the German Federal Ministry of

Education and Research. The funders had no role in study design, data collection and analysis, decision to publish, or preparation of the manuscript.

**Competing interests:** The authors have declared that no competing interests exist.

prevalent in animals [8, 9] and humans [10]. Behavioral experiments further confirm that humans are willing to use costly punishment instruments to establish social norms [11–16], although perhaps not in all circumstances [17–19]. This had led some authors to conjecture that strong reciprocity [20, 21]—a propensity to cooperate and punish defectors, even at a cost to oneself—is a predisposition of human nature and underlies the unprecedented level of human sociality; other mechanisms through which the second-order free-riding problem can be solved were extensively explored in recent years [13, 22–24].

A key question that has received comparably little attention is the role of noise in the evolution of social norms [25]. But human punishment is, in fact, very noisy [26]. Court sentencing, for instance, has long been known to display large inter-judge disparity [27, 28], and judicial errors are common [29]. Perhaps more shockingly, trivial but unpredictable factors appear to have large effects on sentencing: in one study, the rate of favorable decisions dropped from ≃65% to ≃0% as the next food break was approached [30]; in another, sentences were found to be correlated with the outcome of football games, with an average of 136 extra days of jail time assigned to juvenile felons after an upset loss of the local team [31]. Besides raising obvious questions of fairness and equality before the law, these observations put into question the relevance of experimental designs based on strictly deterministic punishment. What is the effect of punishment noise on the evolution of cooperation and maintenance of social norms?

In a typical public goods game (PGG) with punishment, subjects can decide how much to allocate to punish other members of the group after learning their contribution decisions [11, 32]. Fines are then determined by multiplying the amount assigned by the punisher by a punishment enhancement factor, $\beta$, usually taken equal to 3. With two exceptions [33, 34], there is no literature investigating the effect of stochastic $\beta$ on the evolution of social norms of punishment and cooperation; the provisional conclusion appears to be that uncertainty does not affect the likelihood of punishing others, nor the level of cooperation [34].

Our results paint a much different picture. We find that players pay a twofold cost to noise: through lower contribution levels, resulting in lower payoffs; and through increased punishment, including directed towards cooperators. Antisocial punishment in public good games has been observed previously [16, 35, 36], but—being mutually harmful—is difficult to explain in rational terms. We argue that sanctioning noise weakens the foundation on which social norms are erected: if you cannot trust that the other person has control over the things they do, then punishment loses its meaning as an institution, and the possibility of a reliable social contract is destroyed. This may be why antisocial punishment is more prevalent in countries with a weak rule of law [16], where noise is rampant by definition. This conclusion is also in line with experimental results obtained in a complementary setting where players had inaccurate information about others' behavior [35], or with volatile public goods [37].

## Results

### Experimental design

Our experiments replicate the approach of Ref. [16], which in turn follows the highly replicated design of Ref. [11]. 320 participants from 41 countries played an online PGG with punishment in groups of 4 participants. The experiments lasted for 22 rounds (with the first two runs as practice runs), and the subjects were shuffled each round so that their identity remained unknown. See the Methods section below for further details.

Each round consisted of two stages (Fig 1). In the first stage, the subjects played a standard PGG: each was given 10 money units (MU) and could decide how much to contribute to a public project in the group. All contributions were multiplied by an enhancement factor of $r = 2$ and divided equally among group members. After this contribution stage, participants

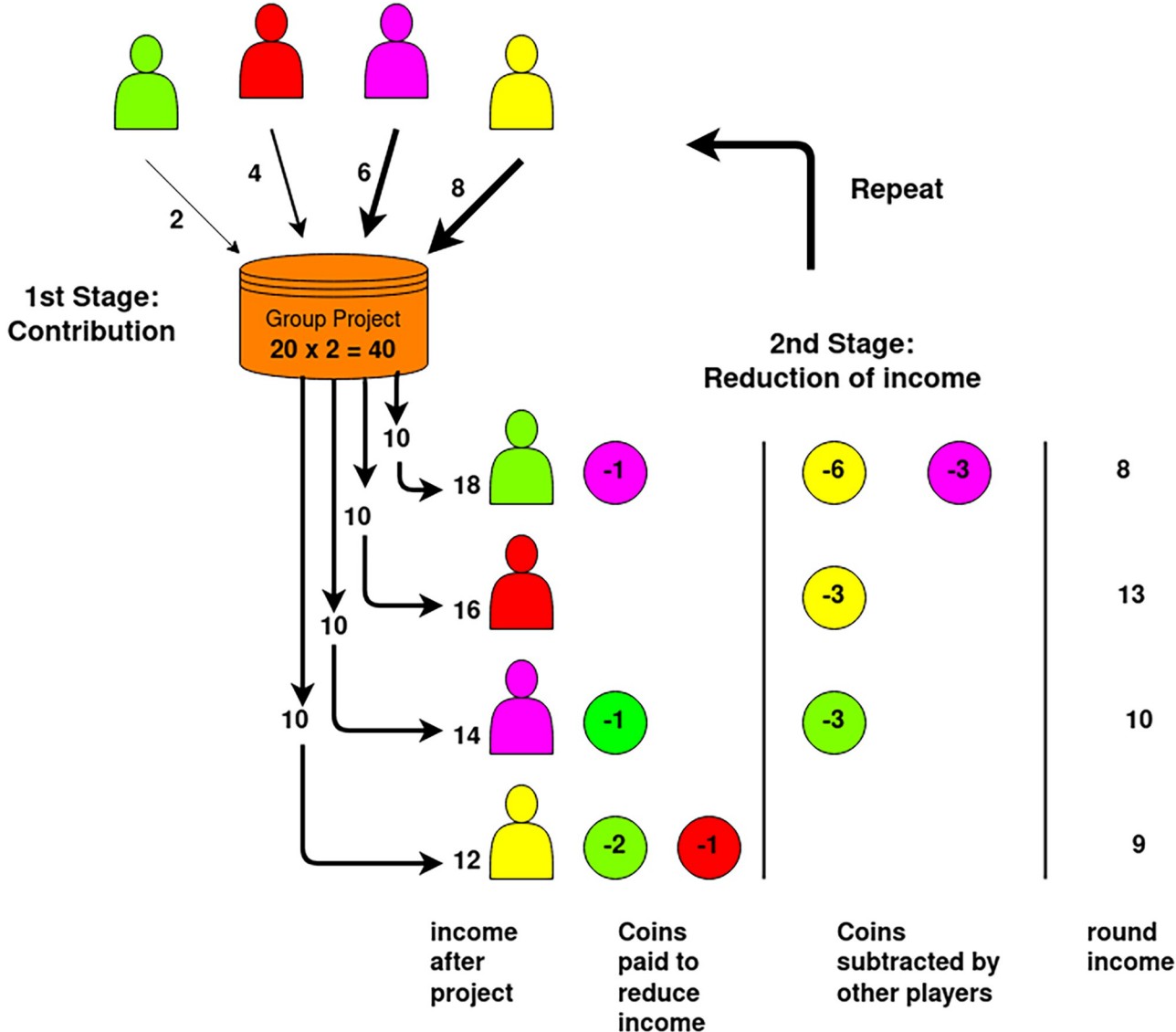

**Fig 1. Experimental design.** Each round of our PGG experiments consists of two stages. In the first stage, participants invest an amount from 0 to 10 MUs in a public good ("group project"). The returns to this investment (the total amount invested times $r = 2$) are then shared equally between all participants, irrespective of their contribution. In the second stage, participants are given an opportunity to reduce the income, i.e. punish, other participants. For this, they can spend up to 10 MUs ("coins") to punish other participants; each MU is multiplied by a factor $\beta$ and the resulting amount is substrated from the punishee's account. In the control group, $\beta$ is fixed to the value 3; in treatment groups, $\beta$ is a random variable with mean 3 and varying standard deviation.

received information about other group members' contributions in this stage and were given an opportunity to sanction other players. To this aim, they could assign up to 10 MU to punish any given player. In the control group, each MU assigned to punish translated into a fine of 3 MU imposed on the punishee. In the treatment groups, punishment was noisy rather than deterministic: instead of the value 3, the punishment enhancement factor $\beta$ was drawn from a uniform distribution with support [2, 4] (low noise), [1, 5] (medium noise), and [0, 6] (high noise). In each case, the average value was equal to 3, *i.e.* there was no bias.

## Group-level outcome: The two-fold cost of noise

We find that noise in punishment is strongly detrimental to cooperation and group welfare in the PGG. As shown in Fig 2, the mean payoff per round accrued by a player over the course of the experiment dropped from an average of 10.1±1.0 MU in the control group to 5.6±1.2 in the high noise group ($p < 0.001$; here and below all $p$-values refer to the two-sided Wilcoxon rank-sum test comparing a treatment to the control). Two main factors underlie this degraded outcome under noise: contributions levels dropped significantly (Fig 3a), and punishment efforts were both more frequent (Fig 3b) and more intense (Fig 3c). The strength of these effects increases monotonically with the strength of the noise. For instance, the probability to punish at least one player rises from 35±4% in the control group to 53±4% in the high noise group ($p = 0.0012$); at the same time, the average cost paid to punish other players rises from 1.95±0.33 to 2.8±0.4 MU ($p = 0.011$).

Underlying lower contributions in noisy conditions was a "hedging" strategy pursued by many players. As illustrated in Fig 4, maximal contributions—common in the control group—were rarely observed under strong noise. Instead, many players chose to contribute about half the maximal amount, $\simeq$5–6 MU. Moreover, while 62±5% of players contributed $\geq$8 MU in the deterministic control, this fraction dropped to 35±5% (resp. 43±5%, 31±5%) in the low (resp. medium, high) noise groups. Thus, instead of full cooperation (which maximizes overall returns) and full defection (the selfish Nash equilibrium), subjects opted for an intermediate strategy of partial cooperation which does not maximize income, be it at the individual level or at the group level.

## Punishment patterns: The rise of antisocial punishment

High levels of punishment do not necessarily imply low payoffs: if wisely used, punishment can increase overall payoffs by enforcing higher contributions. That this is not seen in the treatment groups suggests individuals use punishment inefficiently under noisy conditions. Fig 5 plots the probability and intensity of punishment as a function of the difference between the punishee's and the punisher's contribution to the public good. A negative difference means that the punishee has contributed less than the punisher, and thus punishment is prosocial; a positive contribution difference indicates that the punishee has contributed more than the punisher, and thus punishment is antisocial. In all treatments, strongly prosocial punishment, that is, punishment of players who contribute 8 or more MU below the contribution of the punisher, is less frequent than in the control groups. *Vice versa*, antisocial punishment is more prevalent in all noisy treatments. In total, the average cost paid to punish antisocially almost doubled from 0.42±0.09 MU in the control group to 0.93±0.16 in the high noise group; this increase is significant at the $p = 0.013$ level.

An ordinary least-square regression analysis of group-average contribution, presented in Table 1A, confirms this picture: we find that contribution decreases with noise amplitude ($\sigma = 0, 1, 2, 3$ in respectively, the control, low, medium, and high noise groups) across all the treatments (model 1). However, the dependence of contributions on noise is mediated by a change in punishment patterns. When we control for the group-average prosocial and antisocial punishment (model 2), the association between contribution and noise ceases to be significant. Instead, contributions show a negative association with antisocial punishment, which—as we will show shortly—in turn increases with noise. These two facts—noise increases antisocial punishment, antisocial punishment decreases contributions—explains the degraded outcome observed in the noisy treatments.

These effects are confirmed when we model directly the dependence of payoff on contribution and punishment patterns within the group. As shown in Table 1B, payoffs drop with the

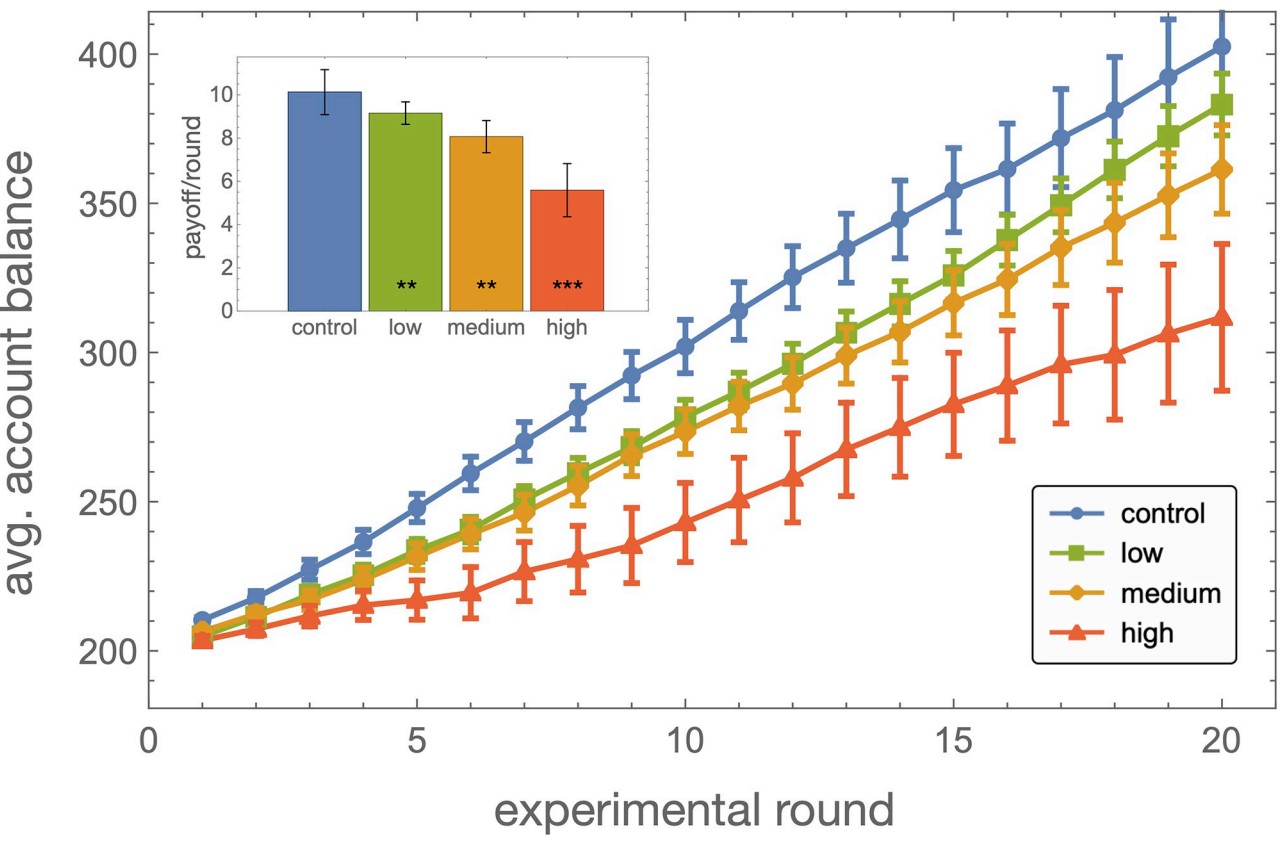

**Fig 2. The cost of noise.** The average account balance during the course of the experiment, for the control (blue), and the low (green), medium (orange), and high (red) noise conditions. Averages are taken over all groups. Inset: Mean payoff per round (the after contribution and punishment stages). Error bars are standard errors, and stars refer to Wilcoxon rank-sum tests comparing each treatment with the control ($p = 0.006$, $p = 0.004$ and $p = 0.0008$ respectively).

noise amplitude (model 1), but this is explained by punishment patterns and initial contribution (model 2); in particular, a group that starts off more generous and punishes less continues to contribute more, and collect higher payoffs.

## Panel analysis: Why punishers punish

To understand how subjects' made their punishment decisions, we performed a censored regression model of individual punishment decisions, considering prosocial and antisocial punishment separately. The dependent variable in these regressions is the number of MU assigned to punish other players, and explanatory variables are the punishee's contribution, the punisher's contribution, the punishment received by the punisher in the previous round, the round number, the average contribution of others (except the punishee and the punisher), and the noise amplitude.

Both antisocial and prosocial punishment decrease with the punishee's contribution, with a stronger effect for prosocial punishment (Table 2). That is, the more a subject contributes, the less they are sanctioned, especially when punishment is prosocial. *Vice versa*, we find that the more a subject contributes, the less they punish others antisocially, and the more they punish others prosocially. These associations are all strongly significant.

In addition, both prosocial and antisocial punishment increase with the number of MU assigned to sanction the punisher by other group members in the previous round. This

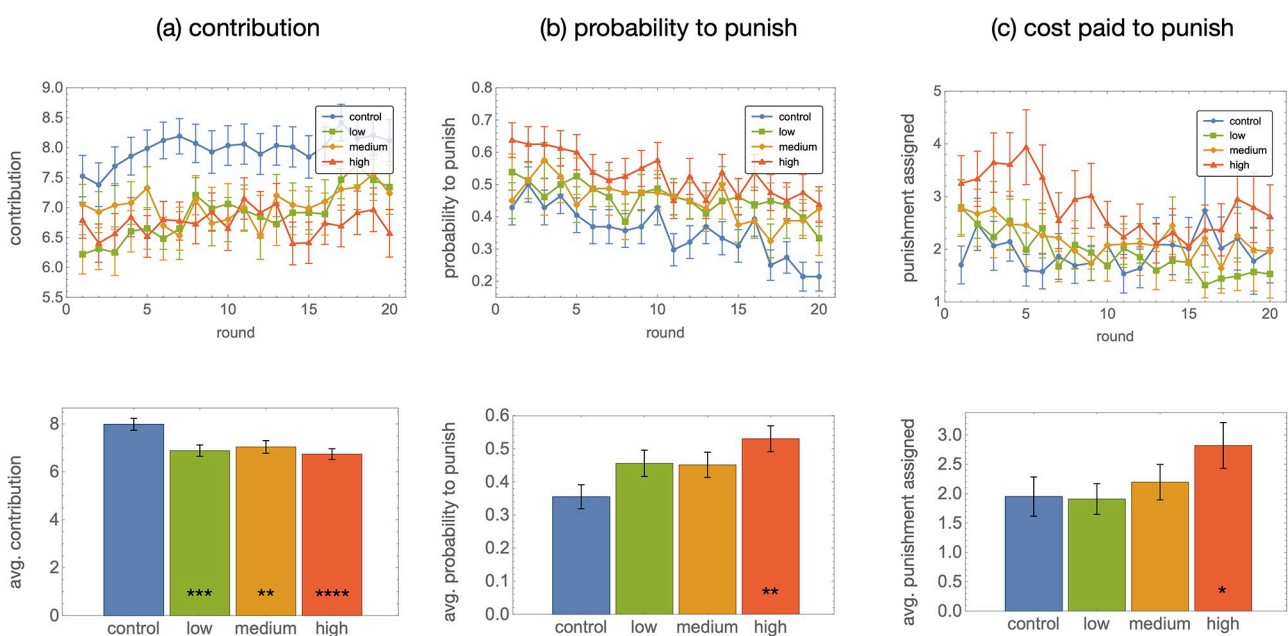

**Fig 3. Lower contributions, stronger punishment.** Time series (top row) and time-averaged (bottom row) contribution to the public good (a), probability to punish at least one player (b), and total cost paid to punish (c). Error bars are standard errors, and stars refer to Wilcoxon rank-sum tests comparing each treatment with the control.

observation suggests revengeful motives are among the reasons why subjects engage in punishing others. Finally, antisocial punishment shows a strongly significant positive relation with noise amplitude. In the case of prosocial punishment, this correlation is much weaker and only marginally significant.

## Discussion

Noise is ubiquitous in the real world, yet we know little about the ways in which it affects social behavior. Here, using online PGG experiments, we showed that noise in punishment outcomes increases punishment efforts, but due to a rise of antisocial punishment, this does not result in stronger cooperation. Social norms are weakened, and individuals lose twice: once to higher destruction of wealth in punishment, and once to lower contributions yielding lower returns. Outside the alternative between a "tragedy of the commons" and a "comedy of the commons", we find that, under uncertain conditions, participants opt for non-committal strategies without any clear economic rationale.

The motivation for humans to engage in antisocial behavior has been subject to much debate. When individuals can benefit from antisocial behavior, for instance when free-riding pays, its occurrence can be simply attributed to self-regarding motives. What is odd about antisocial punishment is that it is a mutually harmful—hence irrational—behavior. Perhaps cultural variation can explain differences in antisocial punishment across societies [16]. According to this view, the prevalence of antisocial punishment in some societies reflects a lack of norms of civic cooperation: antisocial punishment is observed in the lab simply because participants bring their society's norms—a weak rule of law—into the lab. Other explanations for antisocial punishment have been proposed. Thus, norm conformity, the idea that antisocial punishment arises due to humans' tendency to punish atypical behavior, may be another factor

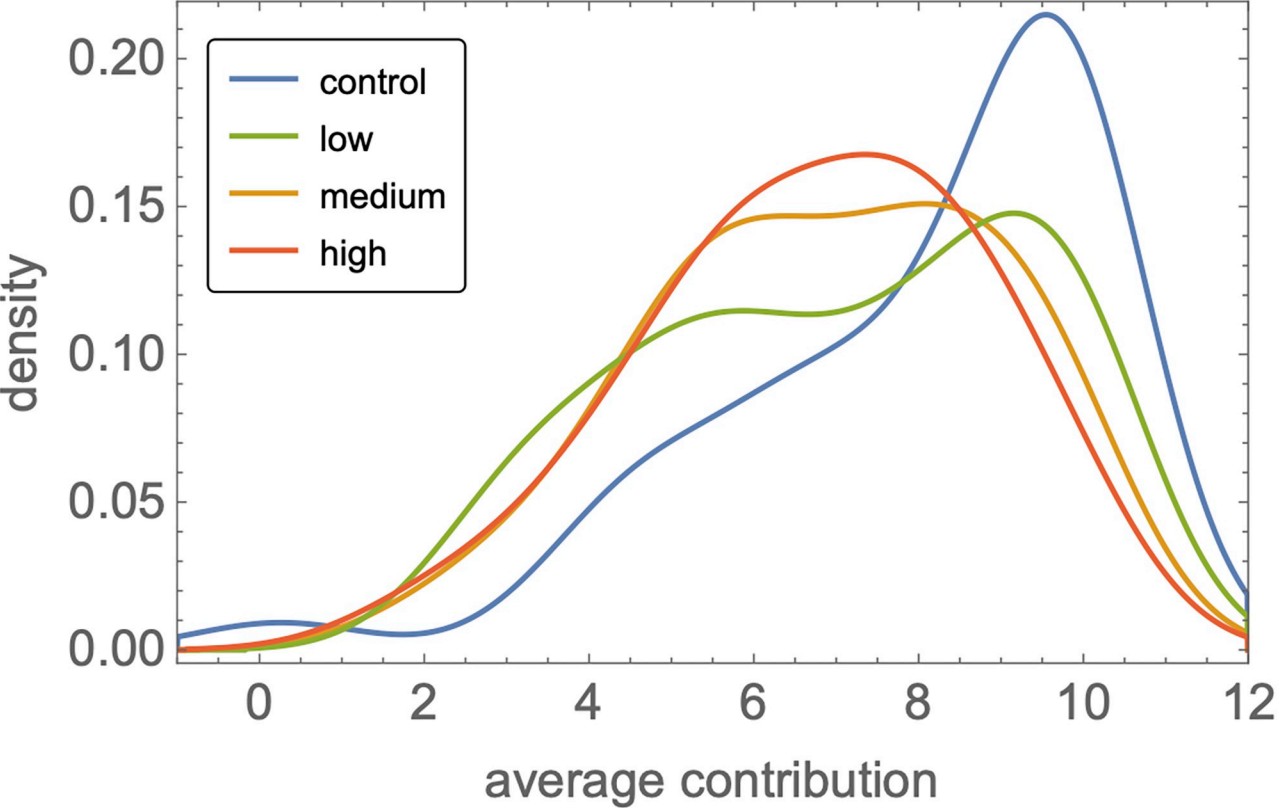

**Fig 4. Hedging one's bets.** Density of per-player average contributions in the four groups. While most players settled on near-maximal contributions in the control group (≃10 MU), noise induced many participants to "hedge their bets" and contribute intermediate amounts (≃5 MU). Densities above 10 or below 0 MU are an artifact of the smoothing procedure.

[38]. Other authors have cited the struggle for status within groups [39], or revengeful motivations [40], as other possible explanations for antisocial punishment.

Our experiments show that uncertainty can be yet another factor at work in the emergence of antisociality. What mechanism underlies an increase in antisocial punishment in noisy conditions? Although our experiments were not designed to address this question, a plausible answer may be that noisy punishment weakens the link between punishment and sociality. Even when individuals with prosocial dispositions try to establish a social norm by punishing defectors, noise in the outcome interferes with that outcome. This can have two effects. First, risk increases in the system. According to prospect theory, actors are risk-averse when confronted with gains, but risk-taking in loss situations [41]. As the threat of punishment constitutes a potential loss to the individual, noise in punishment is expected to lead to risk-taking behavior, hence in this case to lower cooperation. Second, unjustified punishment caused by noise can violate a punishee's perception of fairness, and trigger anger and spite. As argued previously [16], spiteful acts can lead to a self-reinforcing outburst of antisocial punishment: individuals punish irrationally because they feel they are treated unfairly, and this, in turn, triggers more spite. Overall, the conditions for a reliable social contract are not met, and group welfare deteriorates.

An interesting question is how the marginal per capita return (enhancement factor $r$ divided by group size), which measures the weakness of the social dilemma, affects our results [42]. Past experimental works have shown that cooperation in public goods experiments falls

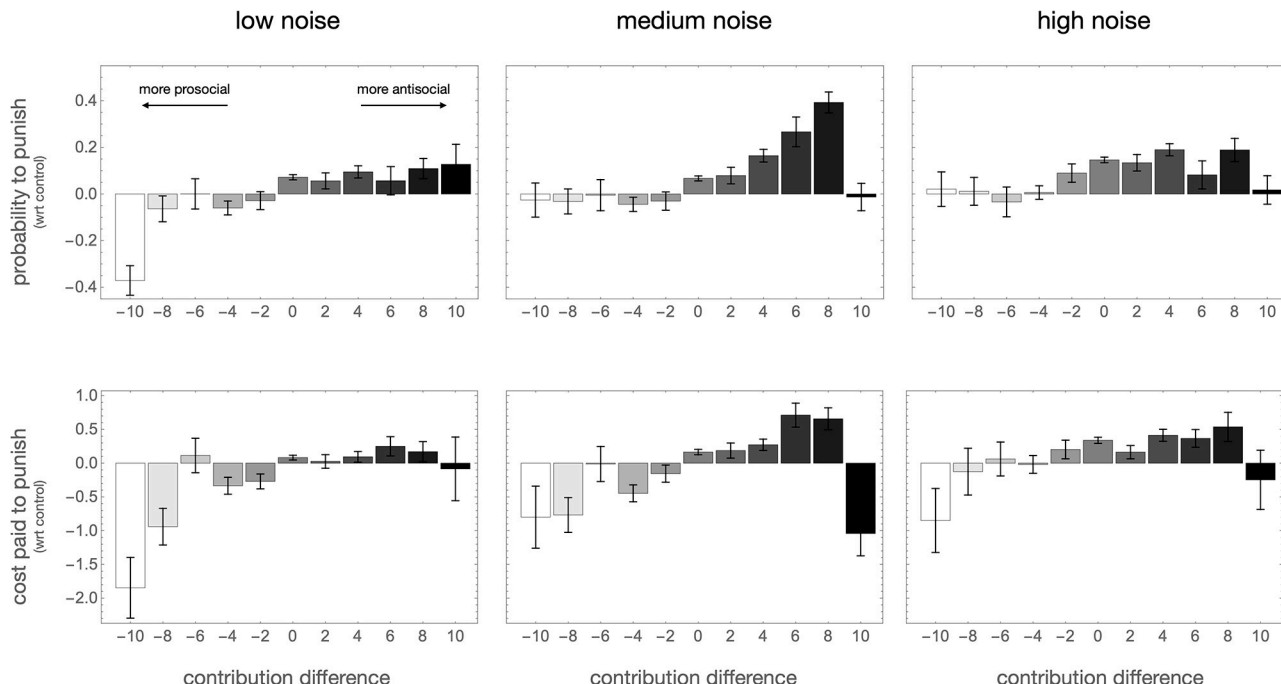

**Fig 5. Uncertainty begets antisociality.** The probability to punish a player (top row) and the average cost assigned to punish them (bottom row) as a function of the contribution difference = (punishee's contribution) − (punisher's contribution), compared to the control. Under noise, the probability and intensity of strongly prosocial punishment (negative contribution difference) are reduced, and the probability and intensity of antisocial punishment (positive contribution difference) are increased.

for stronger social dilemmas strengths [43–45]. We expect that, for the same reason, noise can more easily disrupt the evolution of social norms when the social dilemma is stronger. Future experimental work can address this question. Similarly, we can ask how the deleterious effect of noise on the evolution of social norms correlates with the "punishment technology" (the mean punishment multiplication factor $\beta$). It is known that $\beta$ should be large enough for prosocial punishment to evolve and cooperation to be sustained in public goods experiments [40, 46]. Furthermore, it has been argued theoretically that low $\beta$ tends to elicit antisocial, rather than prosocial, punishment [47]. These considerations suggest that noise can be even more disruptive for the evolution of social norms—and more facilitating for the evolution of antisocial behavior—when punishment is less stringent (low $\beta$). Here too, future empirical work can clarify this question.

Cross-cultural differences in punishment have been a focus of recent research [16, 18], for instance with regard to cross-cultural variations in antisocial punishment [16], or human attitude towards using punishment instruments [18]. An interesting question in this regard is whether the impact of punishment noise on humans' incentives to use punishment instruments is subject to cross-cultural variation. This may be the case due to several reasons. It is argued that the cognitive bases for blame, revenge and the perception of harm are subject to cross-cultural variations [48, 49]; these may play a key role when punishment is contaminated with noise. Similarly, the perception of and reaction to risk are believed to depend on cultural factors [50–52], which may trigger different responses to noisy punishment. Finally, the very predisposition of humans to use sanction to establish social norms is subject to cross-cultural variation [16, 18]. It would be interesting to explore whether such cross-cultural differences are indeed at work in our results.

**Table 1. Ordinary least square regression model for contributions (A) and payoffs (B).** Group average contributions from period 2 to 20 (A) and group average payoffs from period 1 to 20 (B) are the dependent variable. In Model 1, noise amplitude is used as the dependent variable. In model 2, group average contributions in period 1, group average payoffs, group average antisocial and prosocial punishments are added as independent variables. While contributions and payoffs show a decreasing trend with respect to the noise amplitude (Model 1), controlling for other variables (Model 2) can better explain this pattern.

| | A) Dependent variable: Group average contribution | |
| --- | --- | --- |
| | Model 1 | Model 2 |
| Noise amplitude | -0.38567**(0.1762) | -0.2446 (0.1505) |
| Group average contribution in period 1 | - | 0.4194*** (0.0915) |
| Group average prosocial punishment | - | 0.4242 (0.2696) |
| Group average antisocial punishment | - | -0.9011** (0.3797) |
| Constant | 7.7087***(0.3288) | 4.5939*** (0.7266) |
| Observations | 79 | 79 |
| Adjusted $r^2$ | 0.046 | 0.330 |
| F statistics | 4.79 | 10.6 |
| P-value | 0.0317 | 0.0000 |

| | B) Dependent variable: Group average payoff | |
| --- | --- | --- |
| | Model 1 | Model 2 |
| Noise amplitude | −1.511*(0.8049) | -0.5149 (0.3903) |
| Group average contribution in period 1 | - | 0.7655*** 0.2506 |
| Group average prosocial punishment | - | -2.2926*** (0.7384) |
| Group average antisocial punishment | - | -8.5938*** (1.0399) |
| Constant | 10.289***(1.5019) | 11.814*** (1.9898) |
| Observations | 79 | 79 |
| Adjusted $r^2$ | 0.0313 | 0.755 |
| F statistics | 3.52 | 61.2 |
| P-value | 0.0643 | 0.0000 |

Note:

*$p<0.1$;

**$p<0.05$;

***$p<0.01$

Our study has some implications for criminal law. Legal processes can be subject to uncertainty due to different factors, such as judicial error or ambiguity of language [26]. Past research has shown that informational uncertainty and different types of error can have a detrimental effect on social behavior and the effectiveness of sanctions [26, 35, 53, 54]. However, another aspect of uncertainty in legal procedure is the uncertainty of sanctions. While uncertainty in punishment was argued to increase law-abidance [55], our study suggests that uncertainty represses the evolution of social norms. Reducing uncertainty in the legal process may therefore help to sustain social behavior.

## Methods and materials

The experiments were conducted from March to July 2021 using oTree [56]. We recruited participants using the online platform prolific.co. The sole reason for exclusion was prior participation in a study. Participants were paid a fixed compensation of £3.75 with the opportunity to earn a bonus of up to £6 based on their income in the game. In total, 384 participants in 96 groups finished the experiment. After removing groups with repeated non-responses in order

**Table 2. Tobit model for antisocial and prosocial punishment across all the treatments.** The dependent variable is punishment points assigned. A Tobit model where the dependent variable is bounded between zero and 10 is used.

| | Dependent variable: Assigned punishment points | |
| --- | --- | --- |
| | **Antisocial punishment** | **Prosocial punishment** |
| Punishee's contribution | -0.1420*** (0.0292) | -0.4606*** (0.0245) |
| Punisher's contribution | -0.1628*** (0.0251) | 0.0599** (0.0291) |
| Punishment received at $t - 1$ | 0.0315*** (0.0050) | 0.0332*** (0.0057) |
| Round | -0.0624*** (0.0112) | -0.0206** (0.0111) |
| Last round | 0.4796* (0.2966) | -0.2064 (0.2999) |
| Average contribution of others in group | 0.1581*** (0.0246) | 0.3067*** (0.0247) |
| Noise amplitude | 0.4637*** (0.0524) | 0.0743* (0.0513) |
| Constant | -1.4232*** (0.2936) | -1.3921*** (0.2937) |
| Observations | 5995 | 5995 |
| Sigma | 3.514 | 3.777 |
| Log likelihood | -5229 | -6618 |

Note:

* $p < 0.1$;

** $p < 0.05$;

*** $p < 0.01$

to eliminate bias, 80 groups remained for the analysis, involving 320 participants from 41 countries (sex ratio $\simeq 2:1$). These are distributed on the experimental conditions in the following way: 21 groups in the control condition, and 20 (resp. 20, 19) groups in the low (resp. medium, high) noise treatment condition.

After accessing the study via study link, the participants were presented with an information sheet to which they had to agree and were asked to identify themselves using their Prolific identification. After that, participants were shown six pages of instructions explaining the game, followed by a set of control questions to ensure their understanding. The content of the information sheet, the instruction pages, and the control questions are listed below.

After correctly completing the control questions, participants were assorted into groups of four. The identity of the other players was concealed, their displayed order randomized in between rounds. They played 22 rounds of a Public Goods Game, the first two rounds being labeled as test rounds, not counting towards the final income. The actual number of rounds was not communicated to the participants to avoid defective behavior in the final rounds.

At the start of the game, participants were endowed with 200 Coins, with each Coin worth £0.01. The game itself was divided into two stages. In the first stage, participants were given 10 Coins, of which they had the choice to contribute any amount to a non-specified group project and keep the rest for themselves. Each Coin invested in the group project was multiplied by factor 2 and distributed evenly between all group members.

In the second stage, participants were shown the other group members' contributions to the group project and allowed to spend up to 10 Coins per group member to reduce that player's income. The experiment contained four experimental conditions. The treatment conditions differed by the factor used to enhance the amount paid to reduce other players' income. In the control condition, each Coin spent to reduce income was multiplied by three, and the result was subtracted from the punished player's account. The three treatment conditions introduced a noise parameter of varying degrees, with the multiplication factor for each payment being drawn from a continuous uniform distribution for each treatment condition. The

distributions all exhibit a mean of 3 but vary widely in range. Distributions with the bounds 2 to 4, 1 to 5, and 0 to 6 were chosen. After seeing the effects of reduction payments and the amount of punishment received by others, the participants proceeded to the next round. At the end of the game, the final income from the game was shown, and participants were asked to answer a number of survey questions that are listed below. Participants that concluded the survey were given a completion link to receive their compensation and bonus via Prolific.

## Supporting information

**S1 Text. Supplemental information text.** The details of experimental procedure and supplementary analysis of the experimental data is presented.
(PDF)

## Acknowledgments

We thank Bapu Vaitla, Wolfram Barfuss and Ian H. Hatton for useful comments on earlier versions of this manuscript.

## Author Contributions

**Conceptualization:** Mohammad Salahshour, Matteo Smerlak.

**Data curation:** Vincent Oberhauser.

**Formal analysis:** Mohammad Salahshour, Vincent Oberhauser, Matteo Smerlak.

**Funding acquisition:** Matteo Smerlak.

**Investigation:** Mohammad Salahshour, Vincent Oberhauser.

**Supervision:** Matteo Smerlak.

**Validation:** Matteo Smerlak.

**Visualization:** Matteo Smerlak.

**Writing – original draft:** Mohammad Salahshour.

**Writing – review & editing:** Mohammad Salahshour, Vincent Oberhauser, Matteo Smerlak.

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
