## [Decision Letter · Decision Letter 0]

18 Oct 2021

PONE-D-21-31875The cost of noise: stochastic punishment falls short of sustaining cooperation in social dilemma experimentsPLOS ONE

Dear Dr. Smerlak,

Thank you for submitting your manuscript to PLOS ONE. After careful consideration, we feel that it has merit but does not fully meet PLOS ONE’s publication criteria as it currently stands. Therefore, we invite you to submit a revised version of the manuscript that addresses the points raised during the review process.

We look forward to receiving your revised manuscript.

Kind regards,

Jun Tanimoto

Academic Editor

PLOS ONE

Journal Requirements:

Reviewers' comments:

Reviewer's Responses to Questions

**Comments to the Author**

1. Is the manuscript technically sound, and do the data support the conclusions?

Reviewer #1: Yes

Reviewer #2: Yes

2. Has the statistical analysis been performed appropriately and rigorously? 

Reviewer #1: Yes

Reviewer #2: Yes

3. Have the authors made all data underlying the findings in their manuscript fully available?

Reviewer #1: Yes

Reviewer #2: No

4. Is the manuscript presented in an intelligible fashion and written in standard English?

Reviewer #1: Yes

Reviewer #2: Yes

5. Review Comments to the Author

Reviewer #1: This is a substantially informative and admirably interesting work, which highlights on PGG with punishment scheme from experimental approach. The authors main concern is whether noise of punishment process enhancing or devastating cooperation; and entailing pro- or anti-social punishing intention.

Their experimental design was quite intrigued and data analysis in view of statistics was scientifically solid. Visual results are impressive and persuasive, which prove that noisy punishment deteriorates the bolster effect by a punishment scheme. I’m impressed their finding that, despite an individual who punishes defectors from his fair intention in terms of a social norm, a noisy punishment system, more or less, inevitably affects on the result.

Since visuals and text aside the content itself are all well-organized, I would like to suggest this MS as it is to publication. Yet, I would give several comments as below, which are expected to be considered to finalize their MS.

One this is their experimental design. PGG presumes the so-called dilemma weakness parameter; r of 2 while the average amplification factor of punishment (that is denoted by the average ration of fine; Beta, and cost; Alpha) is 3. I wonder why the authors designed (chosen) this particular setting; (r, Ave[Beta/Alpha]) = (2, 3). I definitely believe their result would have a keen sensitivity from the combination of (r, Ave[Beta/Alpha]). Or simply, what happens if bit more severe dilemma situation (less than r = 2) is levied to responders?

As well known, PGG can be translated to one of the sub-classes of 2 by 2 Prisoner’s Dilemma; Donor & Recipient (D & R) game (reference, e.g.; Difference of reciprocity effect in two coevolutionary models of presumed two-player and multi-player games, Physical Review E 87, 062136, 2013). And if one references to the new concept of Social Efficiency Deficit; SED (reference, e.g.; Social efficiency deficit deciphers social dilemmas, Scientific Reports 10, 16092, 2020), under a bit tough dilemma situation, human’s intention or expectation to the scheme of Punishment so as to maintain the social fairness would be altered. Of course, the authors’ result from the current experiment could tell noting about this point. Yet, I really love to hear the authors opinion on the point in Discussion part or their provision on future works. When some comment and discussion would be added, the relevant literature as above should be cited.

Reviewer #2: Following the experimental design of Science 319.5868 (2008), and Am. Econ. Rev. 90.4 (2000), the authors studied the influence of noisy punishment to the evolution of cooperation. They found that the contributions decrease and punishment efforts intensify with punishment noise increases. Besides, they also observe that uncertainty causes a rise in antisocial punishment. Totally speaking, they authors do a good job, their results and analysis are beautiful and impressive. I support its acceptance before the authors address the following questions.

1. Page 3, Experimental design, the authors say "320 participants from 41 countries played online PGG with punishment in groups of 4 participants". I just curious about whether there are any cross-culture effect regarding to the uncertainty causes a rise in antisocial punishment. As the authors cited in the manuscript, ref. 16 and ref. 18, cultural difference or social norm plays an important role to sustaining cooperation. Since the authors already have data from different countries, I suggest the authors to investigate whether there is a cross-cultural effect regarding the influence of noisy punishment, or the necessary discussions are needed.

2. Ref 17 and Ref 18 were with same experimental conditions but they were conducted on different countries, the results are totally opposite, what's the behind the reason for this opposite results, is it related with noise? If possible, please write your considerations in Discussion part.

6. PLOS authors have the option to publish the peer review history of their article (what does this mean?). If published, this will include your full peer review and any attached files.

Reviewer #1: No

Reviewer #2: No

---

## [Author Response · Author response to Decision Letter 0]

19 Nov 2021

Please see attached "RevieweResponse.pdf"

---

## [Decision Letter · Decision Letter 1]

11 Jan 2022

The cost of noise: stochastic punishment falls short of sustaining cooperation in social dilemma experiments

PONE-D-21-31875R1

Dear Dr. Smerlak,

We’re pleased to inform you that your manuscript has been judged scientifically suitable for publication and will be formally accepted for publication once it meets all outstanding technical requirements.

Kind regards,

Jun Tanimoto

Academic Editor

PLOS ONE

Additional Editor Comments (optional):

Reviewers' comments:

Reviewer's Responses to Questions

**Comments to the Author**

1. If the authors have adequately addressed your comments raised in a previous round of review and you feel that this manuscript is now acceptable for publication, you may indicate that here to bypass the “Comments to the Author” section, enter your conflict of interest statement in the “Confidential to Editor” section, and submit your "Accept" recommendation.

Reviewer #1: All comments have been addressed

Reviewer #2: (No Response)

2. Is the manuscript technically sound, and do the data support the conclusions?

Reviewer #1: Yes

Reviewer #2: Yes

3. Has the statistical analysis been performed appropriately and rigorously? 

Reviewer #1: Yes

Reviewer #2: Yes

4. Have the authors made all data underlying the findings in their manuscript fully available?

Reviewer #1: Yes

Reviewer #2: Yes

5. Is the manuscript presented in an intelligible fashion and written in standard English?

Reviewer #1: Yes

Reviewer #2: Yes

6. Review Comments to the Author

Reviewer #1: The authors did response the suggestions the reviewers gave. Therefore, I can suggest now the revised MS to be accepted .

Reviewer #2: (No Response)

7. PLOS authors have the option to publish the peer review history of their article (what does this mean?). If published, this will include your full peer review and any attached files.

Reviewer #1: No

Reviewer #2: No

---

## [Editor Report · Acceptance letter]

3 Feb 2022

PONE-D-21-31875R1 

The cost of noise: stochastic punishment falls short of sustaining cooperation in social dilemma experiments 

Dear Dr. Smerlak:

I'm pleased to inform you that your manuscript has been deemed suitable for publication in PLOS ONE. Congratulations! Your manuscript is now with our production department. 

Kind regards, 

on behalf of

Prof. Jun Tanimoto 

Academic Editor

PLOS ONE